# Composite Film Based on Pulping Industry Waste and Chitosan for Food Packaging

**DOI:** 10.3390/ma11112264

**Published:** 2018-11-13

**Authors:** Ji-Dong Xu, Ya-Shuai Niu, Pan-Pan Yue, Ya-Jie Hu, Jing Bian, Ming-Fei Li, Feng Peng, Run-Cang Sun

**Affiliations:** Beijing Key Laboratory of Lignocellulosic Chemistry, Beijing Forestry University, Beijing 100083, China; xujidong@bjfu.edu.cn (J.-D.X.); 18813076766@163.com (Y.-S.N.); ypp1109@bjfu.edu.cn (P.-P.Y.); huyajie0311@163.com (Y.-J.H.); bianjing31@bjfu.edu.cn (J.B.); limingfei@bjfu.edu.cn (M.-F.L.); rcsun3@bjfu.edu.cn (R.-C.S.)

**Keywords:** hemicelluloses, chitosan, composite films, oxygen barrier property, food packaging

## Abstract

Wood auto-hydrolysates (WAH) are obtained in the pulping process by the hydrothermal extraction, which contains lots of hemicelluloses and slight lignin. WAH and chitosan (CS) were introduced into this study to construct WAH-based films by the casting method. The FT-IR results revealed the crosslinking interaction between WAH and CS due to the Millard reaction. The morphology, transmittance, thermal properties and mechanical properties of composite WAH/CS films were investigated. As the results showed, the tensile strength, light transmittances and thermal stability of the WAH-based composite films increased with the increment of WAH/CS content ratio. In addition, the results of oxygen transfer rate (OTR) and water vapor permeability (WVP) suggested that the OTR and WVP values of the films decreased due to the addition of CS. The maximum value of tensile strengths of the composite films achieved 71.2 MPa and the OTR of the films was low as 0.16 cm^3^·μm·m^−2^·24 h^−1^·kPa^−1^, these properties are better than those of other hemicelluloses composite films. These results suggested that the barrier composite films based on WAH and CS will become attractive in the food packaging application for great mechanical properties, good transmittance and low oxygen transfer rate.

## 1. Introduction

The utilization of potentially renewable materials is becoming an increasingly acknowledged and promising alternative for future materials products in the sustainable and green society. Over the past decades, the dominating materials of the food packaging are derived from the non-degradable fossil fuels. However, the films produced from fossil fuels have brought much intricate threats to our environment. Meanwhile, the storage volume of fossil fuels was sharply decreased. Therefore, optimized utilization of renewable biomass has attracted more attention in food packaging application [1,2]. Among the biomass polymers, lignocellulosic biomass is a valuable and uniquely sustainable resource because it could be converted into chemicals, polymeric materials and bioproducts.

The lignocellulosic biomass has complex structure consisting of three main components including cellulose, hemicelluloses and lignin. Hemicelluloses are the second abundant polysaccharides in nature. Hemicelluloses demonstrate many valuable properties, such as excellent biodegradability, biodegradability and remarkable film-forming properties [3,4,5]. Recently, the hemicelluloses based composite films have received increasing concern, especially the application of the films in the food packaging [6]. However, isolated and highly purified hemicelluloses are usually obtained from being extracted by alkaline from the lignocellulosic resources. The alkaline extraction is to add alkaline solution into the lignocellulosic resources and then followed by a series of filtration steps to remove hemicelluloses and some lignin from lignocellulosic resources [7]. In addition, the purification process of hemicelluloses is complex and economically infeasible. Therefore, the forming of hemicelluloses-based films would cost much more energy and time. Wood auto-hydrolysate (WAH) is often removed as the wastewater in the pulping process, which is obtained after hydrothermal treatment the wood chips. In the terms of implementation and commercialization, hemicelluloses-rich wood auto-hydrolysate would be the better alternation to form the films, which could be used in the field of food packaging.

Recently, hemicelluloses-rich WAH is shown to be a feasible resource for the design of films. The dominating polymers of WAH are hemicelluloses, lignin, oligosaccharide and monosaccharide. In the previous work, the films based on the macromolecular hemicelluloses with high purity separated from WAH have some great properties, such as good barriers to oxygen, low cost and easy availability [8]. However, the hemicelluloses based composite films exhibit poor mechanical strength, hygroscopic, poor transparency. These drawbacks of the hemicelluloses based composite films limit the practical applications. To improve the performance of these composite films, plasticizers (such as chitosan [9], carboxymethyl cellulose, [10] and xylitol [11]) are often introduced to improve the mechanical strength of the WAH based composite films. In general, macromolecular hemicelluloses were isolated from the WAH by fractional purification methods such as ethanol precipitation, membrane separation and so forth. This study was to prepare composite films using WAH directly instead of using macromolecular hemicelluloses from the WAH. Therefore, this study aimed at the preparation of the composite films based on WAH and CS with the different ratios in volume. In this work, the components and molecular weight of WAH were determined, the morphology, mechanical properties, thermal stability and water vapor permeability of the composite films were also characterized for the further application.

## 2. Materials and Methods

### 2.1. Materials

WAH used in this study, obtained from Eucalyptus wood chips, was provided by Shandong Sun Paper Industry Joint Stock Co., Ltd., Jining, China. Chitosan (CS) was supplied by Sinopharm Chemical Reagent Co. (Shanghai, China), with a medium viscosity of 50–800 mPa·s (CAS 9012-76-4).

### 2.2. Characterization of WAH

The main hemicelluloses were extracted from WAH by ethanol precipitation. Molecular weight of WAH and the extracted hemicelluloses were measured by Gel permeation chromatography (GPC). [12] The high performance anion exchange chromatography (HPAEC) was applied to determine the sugar components of WAH and the extracted hemicelluloses [12]. The acid insoluble lignin of WAH was analyzed by determining gravimetrically and the acid soluble lignin of WAH was determined by the National Renewable Energy Laboratory (NREL) method [13].

### 2.3. Preparation of WAH/CS Composite Films

The composite films were prepared from the blended solutions, which consisted of WAH and CS. The forming mechanism of composite films was the result of the Millard reaction, which existed in the carbonyl of WAH and the amino of CS [14,15]. The WAH (2 wt %) was firstly prepared under stirring for 2 h. The CS solution (2 wt %) was prepared under stirring after adding 1% (*v*/*v*) acetic acid and the obtained CS solution was centrifuged to remove microbubbles. Then the prepared WAH solution and CS solution were mixed and stirred vigorously for 6 h at room temperature. After absolutely dissolved, each aliquot of 10 mL blended solutions was cast into the 60 mm diameter plastic Petri dish and then all the blended solutions were left to be dried at 50 °C in a vacuum drying chamber. The composite films were obtained after being dried about 5 h and easily peeled from the Petri dishes. As shown in Table 1, WAH (2 wt %) and were blended with 2 wt % CS in different volume ratios and the forming pathway of the composite films WAH/CS is illustrated in Figure 1.

### 2.4. Characterization of WAH/CS Composite Films

The surface and cross-section morphology of the composite films based on WAH and CS were analyzed by SEM with the instrument of Hitachi S-3400N II (Hitachi, Tokyo, Japan). Firstly, the composite films were sprayed with gold and sent into the instrument. Then, the SEM images at different magnifications (from 200× to 5000×) were obtained. The atomic force microscopy (AFM; Bruker, Germany) images of composite films were used to evaluate the morphology of the surface structure of composite films based on WAH and CS. After gluing the composite films onto metal disks, attaching it to the magnetic sample holder and placing it on the top of scanner tube, the AFM images of composite films were gathered by using a monolithic silicon tip at room temperature. The FT-IR spectra were recorded on a FT-IR Microscope (Thermo Scientific Nicolet In 10, Thermo Electron Scientific Instruments LLC, Madison, WI, USA). The FT-IR spectra of WAH/CS composite films were recorded ranging from 4000 to 650 cm^−1^ at a distinguish ability of wavenumber 4 and 128 cm^−1^ scans.

### 2.5. Measurement of Thickness

The thickness values of films were measured on a paper thickness gauge (ZH-4, Changchun paper testing machine CO. Ltd., Changchun, China). The indication of the paper thickness gauge furnishes a pinpoint scale with 0.001 mm resolution. The results of all composite films were based on at least 5 sets of data.

### 2.6. Light Transmittance

The transparency of composite films based on WAH and CS was performed on the UV-Vis spectrophotometer. The cuvettes, the accessory instrument of UV-Vis, were used as the loading gear to load the films WAH-CS. The composite films were cut into be rectangular specimens and then put into the cuvettes for the analysis of the transparencies of the films WAH-CS. The values of transmittance were recorded based on at least 3 sets of data and the corresponding transparencies curves were obtained.

### 2.7. Tensile Strength Testing

The tensile strength of composite films was measured on an Instron 5566 with Bluehill 2 software. The test was carried out at 50% relative humidity (RH), a stabilized extension rate at 5 mm·min^−1^ and a measure length of 40 mm with a load cell of 100 N volume [16] The composite films were cut into rectangular specimens with the width of 10 mm, afterwards kept in store at room temperature in cabinet containing Mg(NO_3_)_2_ solution for at least 3 days. The results of tensile strength were recorded at least 4 specimens.

### 2.8. Thermal Behavior Analysis

The TGA was carried out on a Mettler Toledo TGA/DSC 851 instrument (Mettler Toledo, Columbus, OH, USA) under a nitrogen atmosphere. The 10 ± 0.5 mg Samples were decomposed on aceramic cup. The weight loss was recorded at the temperature ranging from 40 to 700 °C at a 10 °C·min^−1^ ramp. The samples of 5 ± 0.5 mg were loaded into the sealed aluminum cups with matched punctured lids and heated from 35 to 700 °C at a heating ramp of 10 °C·min^−1^.

### 2.9. Oxygen Transfer Rate

According to GB/T1038-2000, the oxygen transfer rate of the WAH/CS composite films were performed on a VAC-V1 differential pressure method of gas permeation apparatus, controlled by the OX2/230 OTR test system. The superficial area of each composite films was 5.0 cm^2^ and the OTR tests were carried out at 23 °C for 24 h under the oxygen atmosphere and the relative humidity (RH) was 50%. The thicknesses of WAH/CS composite films were tested by the paper thickness gauge and the display value of the instrument offered a pinpoint scale of 0.001 mm. The results are based on at least 3 specimens.

### 2.10. Water Vapor Permeability

Water Vapor Permeability of the WAH/CS composite films was determined in accordance with the standard ASTM E 96/E 96M [16]. Each aluminum cup, the loading tools of wet-cup tests, contained 25 g of anhydrous calcium chloride as the desiccant, while the desiccant was dried at 150 °C for 5 h. Then, composite films based on WAH and CS were covered the cups at 23 °C and the cups were put into a cabinet containing water and weighed by a scale of 0.001 g every 1 h. The results are based on at least 3 specimens. The WVP of the composite films were calculated according to the following equation:(1)$$\mathrm{WVP}=\frac{\mathrm{film}\;\mathrm{thickness}\;\left(\mathrm{mm}\right)\times \mathrm{weight}\;\mathrm{augmenter}\;\left(g\right)}{\mathrm{effective}\;\mathrm{area}\;\left({\mathrm{cm}}^{2}\right)\times \mathrm{time}\left(s\right)\times \Delta P}$$
ΔP is the difference value in water vapor pressure across the composite films (23.76 mmHg).

## 3. Results and Discussion

### 3.1. Components of Wood Auto-Hydrolysate

In this work, the components of wood auto-hydrolysate (WAH) were mainly 61.3% hemicelluloses, 10.5% lignin, 12.8% monosaccharide, 13.7% oligosaccharide and 1.65% other insoluble materials of dry WAH. WAH exhibited the molecular weight as follows: *M*_w_ of 2300 g·mol^−1^, *M_n_* of 260 g·mol^−1^ and a polydispersity of 8.8. The hemicelluloses were precipitated by adding three volume of ethanol from WAH. The sugar component of the extracted hemicelluloses is mainly 71.8% xylose, 10.5% glucuronic acid, 7.6% glucose, 7.6% galactose, 1.9% rhamnose and 0.6% arabinose. Based on the sugar composition of the hemicelluloses, the hemicelluloses are mainly composed of glucuronoxylans.

### 3.2. Structural Analysis of WAH/CS Composite Films

The structural analysis of WAH/CS composite films was characterized by using FT-IR measurement. Figure 2a shows the FT-IR spectra of WAH and CS. The characteristic absorption bands of chitosan observed at 1650 cm^−1^ is assigned to –NH_2_ [17]. The signals at 1620 cm^−1^ is related to the 4-*O*-methyl-glucuronic acid or glucuronic acid carboxylate of WAH [18,19]. The signal at 1731 cm^−1^ is attributed to C=O stretching of acetyl groups in the WAH. The prominent band at 1035 cm^−1^ is attributed to the C–O–C stretching vibration of glycosidic linkages, which is the representative peak of xylans [20]. The absorption at around 890 cm^−1^ is due to the carbohydrate C–H deformation, which is characteristic β-glycosidic linkage between the sugar units [21]. As shown in Figure 2b, the spectral profiles and peaks of all the bands are extremely similar, indicating that the films prepared from the mixture of WAH and CS in different volume ratios had similar structure. Compared with the Figure 2a,b, the absorption peaks at 1650 cm^−1^ of CS and 1620 cm^−1^ of WAH disappeared and the new bands generated at 1559 cm^−1^ and 1716 cm^−1^, which suggest that the Millard reaction (C=N double band) occurred between the reducing end of WAH and the amino groups of CS [22].

### 3.3. Morphology of WAH/CS Composite Films

The homogeneity and topography of WAH/CS composite films were observed by SEM and AFM. The SEM images of the composite films based on WAH and CS are presented in Figure 3. As can be seen from Figure 3a,c,e, the surface of composite films based on WAH and CS are smooth and homogeneous with some irregularities, which are due to the man-made destruction to the films based on WAH and CS. It suggested that WAH and CS were diffused evenly in the composite films. The cross-section images of the films are shown in the Figure 3b,d,e. As can be seen, the cross-sections of composite films based on WAH and CS became rougher when the WAH/CS content ratio changed from 3:2 (F_3-2_) to 1:1 (F_1-1_) and to 2:3 (F_2-3_). It might be due to the increment of the reaction intensities of the Millard reaction between WAH and CS in the composite films F_1-1_, the network structure of film F_1-1_ were more compact and tighter, thus leading to the rougher cross-section. When the CS content continues to increase, excess CS probably increase the viscosity of the film F_2-3_, making the cross-section of film F_2-3_ much rougher and obtaining the higher tensile strain and stress strength (Figure 6).

The surface structural analyses of WAH/CS composite films were performed by the AFM, as shown in Figure 4. As can be seen, the surfaces of films were smooth. The root-mean-square (RMS) roughness of the film F_1-1_ was 8.2 nm, which suggested that WAH/CS composite films had smooth surfaces. This is consistent with the SEM results and the smooth surfaces of the films were beneficial to the application of composite films in food packaging materials.

### 3.4. UV-Vis Transmittance of WAH/CS Composite Films

In general, the optical transparencies of composite films reflect the homogeneity of the structure and the miscibility of composite films. The transmittances at wavelength of 200–800 nm and the photograph of composite films are shown in Figure 5a,b respectively. The transmittances of all the composite films under the 800 nm wavelength were above 70%, as shown in Figure 5a, which proved the excellent transparency of WAH/CS composite films. The carboxymethylxylan film were prepared by Alekhina et al. [23], which were highly transparent with a transmittance of 92%. The reason for difference is that the WAH contained some lignin content. In addition, the light transparencies of WAH/CS composite films increased with the increment of wavelength. As can be seen from Figure 5b, composite films based on WAH and CS were at semitransparent and the transparency of the films increased with the increment of WAH/CS content in the films, that is, the relatively high content of WAH is conducive to the transparency of the WAH/CS composite films.

### 3.5. Mechanical Properties of WAH/CS Composite Films

In order to ensure the obtained composite films have adequate mechanical properties, the tensile testing is essential to the composite films. The tensile stress, tensile strain at break and thickness of the composite films were summarized in Table 2. The stress-strain curves of composite films are shown in Figure 6. As can be seen, the tensile strengths of F_3-2_, F_1-1_, F_2-3_ and F_1-4_ were 28.2, 51.5, 67.5 and 71.2 MPa, respectively. The tensile strength and the tensile strain at break of composite films were improved with the increasing of the CS content. It might be due to the increment of the reaction intensities of WAH and CS with the increment of WAH/CS content ratio from 1:4 to 2:3 to 1:1 and to 3:2. The 100% WH films are very brittle, and so fragile that it cannot be tested; this result is consistent with the xylan-based film reported by Gröndahl et al. [24]. However, the composite film F_1-4_ had an astonishingly higher tensile strength, which is indicated by the tensile strain of 6.1% and stress strength of 71.2 MPa. It might be due to the high viscosity of the unreacted CS which improved the tensile strength of the films. In addition, as can be seen from Table 2, compared with the films reported in literatures [15,25,26], the tensile strength of the WAH/CS films were higher than that of the films based on pure xylan or chitosan. Therefore, the films based on WAH and CS are suitable for the application of food packaging with great mechanical properties.

### 3.6. Thermal Behavior of WAH/CS Composite Films

The thermal behavior of composite films based on WAH and CS were investigated by thermogravimetric analysis (TGA). In Figure 7a, the initial weight losses of about 6% are ascribed to the evaporation and release of water. All the composite films started to decompose at around 200 °C. And the weight losses of WAH/CS composite films mainly occurred at the temperature range of 200–700 °C, which was due to the degradation of polymers (WAH and CS), such as the glycosidic bonds and C–O band. Additionally, the *T*_onset_ (the initial degradation temperature), *T*_max_ (the maximum weight loss temperatures) and the residual values of WAH/CS composite films were determined by the DTG curves and all values are shown in Table 3. In Figure 7b, slight differences in *T*_onset_ and *T*_max_ were obviously observed in the DTG curves of the films. As can be seen, the *T*_onset_ of F_1-1_ and F_1-4_ were 204.2 and 237.6 °C and that of F_4-1_ was 183.4 °C. Therefore, the enhancement of thermal stability may be due to the reaction intensities of WAH and CS increased with the increment of WAH/CS content ratio from 1:4 to 1:1 and to 4:1. It was found that there was a slight shift in *T*_max_ during the thermal analysis of composite films. The *T*_max_ of F_4-1_ and F_1-4_ were found at 284.7 and 276.4 °C and that of F_1-1_ was 270.4 °C. In addition, more solid residues were remained in film F_1-1_ than other films at 700 °C (Table 3), which was due to the much stronger interaction between WAH and CS. However, the DTG curve of F_4-1_ had two *T*_max_ values, which might be due to the excess content of WAH. This result is consistent with the FT-IR results.

### 3.7. Permeability Analysis of WAH/CS Composite Films

Oxygen transfer rate (OTR) and water vapor permeability (WVP) should be as low as possible in order to optimize the applications of composite films in food packaging. The results of OTR and WVP of the films based on WAH and CS are summarized in Table 4. As can be seen, the OTR values of composite films F_3-2_, F_1-1_, F_2-3_ and F_1-4_ were 0.34, 0.16, 0.30 and 0.37 cm^3^·m^−2^·24 h^−1^·kPa^−1^, respectively. The OTR value of F_1-1_ was the lowest among the composite films, which was due to the stronger interactions between WAH and CS. The stronger interactions introduced a barrier of the oxygen molecules. The composite films based on WAH and CS were relatively lower than those of the films in the literatures [27,28,29]. As reported in literatures, the OTR values of acetylated galactoglucomannans (AcGGM) film and polylactic acid film are 1.28 and 18.65 cm^3^·m^−2^·24h^−1^·kPa^−1^, respectively. In addition, the standard maximum OTR value of food packaging materials is below 10 cm^3^·m^−2^·24 h^−1^·kPa^−1^ [29]. As shown in Table 4, the WVP values of the composite films F_3-2_, F_1-1_, F_2-3_ and F_1-4_ were 2.42, 2.17, 2.28 and 3.82 (× 10^−10^g·cm·cm^−2^·s^−1^·mmHg^−1^), respectively. The WVP value of the films F_1-1_ was much lower than those of the film F_3-2_, F_2-3_ and F_1-4_, which might be due to the increment of the stronger reaction between WAH and CS. The low WVP value of the composite films is an essential property for the food packaging materials. Therefore, the excellent OTR and WAH made the WAH/CS films more suitable for the application in the food packaging.

## 4. Conclusions

An easy and rapid way was adopted for preparation the barrier films based on WAH and CS was studied in this works. FT-IR analysis suggested that the obtained composite films was the result of the crosslinking interaction between WAH and CS, which is arose from the Millard reaction of the carbonyl of WAH and the amino of CS. The SEM and AFM images suggested the composite films showed a smooth surface and a dense structure. The physical properties of the composite films with different ratio of WAH and CS were also studied. As the analysis revealed, the tensile strength and oxygen barrier ability of the composite films were improved due to the addition of CS. The films based on WAH and CS showed high tensile strength (71.2 MPa), good thermal stability, high transmittances, low water vapor permeability and excellent oxygen barrier properties (<1 cm^3^·m^−2^·24 h^−1^·kPa^−1^), these properties are beneficial to constructing packaging materials. Therefore, composite films based on WAH and CS would become attractive in the application of packaging materials in the food packaging. In summary, converting wood auto-hydrolysate into value-added films could lower the production cost, benefit environment and increase revenue for paper making industry.

## Figures and Tables

**Figure 1 materials-11-02264-f001:**
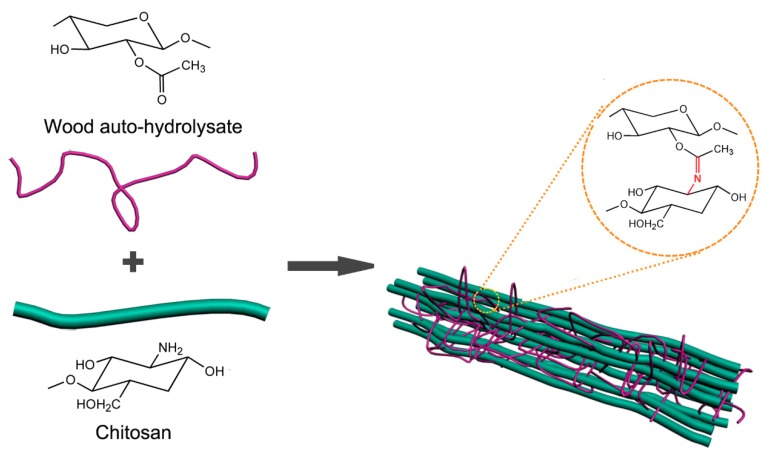
The forming pathway of composite films based wood auto-hydrolysates (WAH) and chitosan (CS).

**Figure 2 materials-11-02264-f002:**
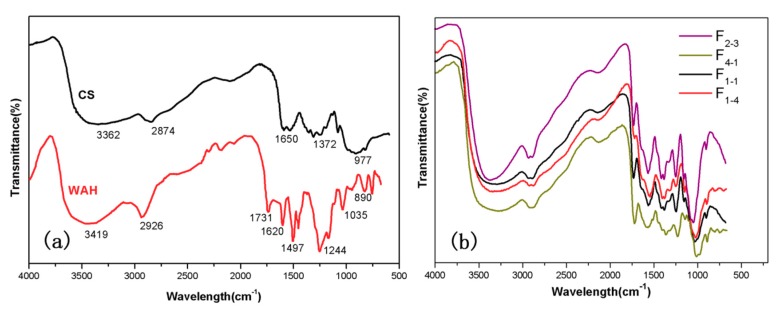
(**a**) Fourier transform-infrared (FT-IR) spectra of WAH and CS, (**b**) FT-IR spectra of represent composite films (F_4-1_, F_3-2_, F_1-1_, F_1-4_).

**Figure 3 materials-11-02264-f003:**
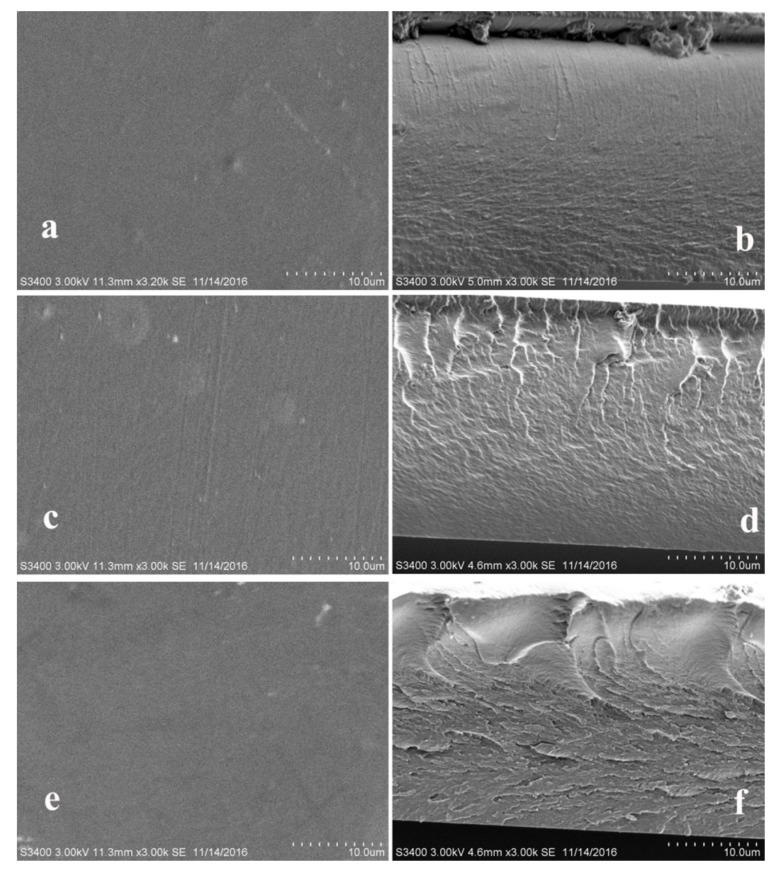
Scanning electron microscope (SEM) images of representative composite films prepared from WAH and CS. (**a**,**c**,**e**) Surface of F_3-2_, F_1-1_, F_2-3_; (**b**,**d**,**f**) cross-section of F_3-2_, F_1-1_, F_2-3_.

**Figure 4 materials-11-02264-f004:**
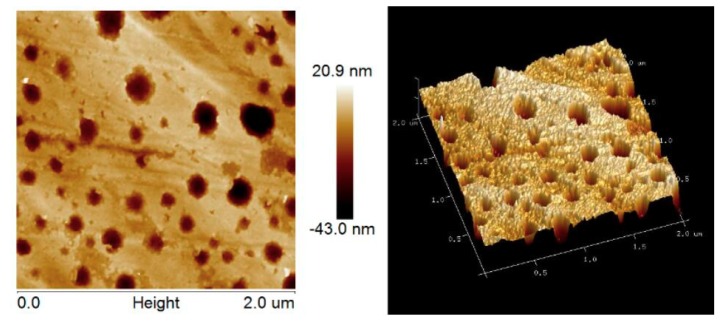
Atomic force microscopy (AFM) images of film, phase image and 3D images of the film F_1-1_ (The scanning scale is 2 × 2 μm).

**Figure 5 materials-11-02264-f005:**
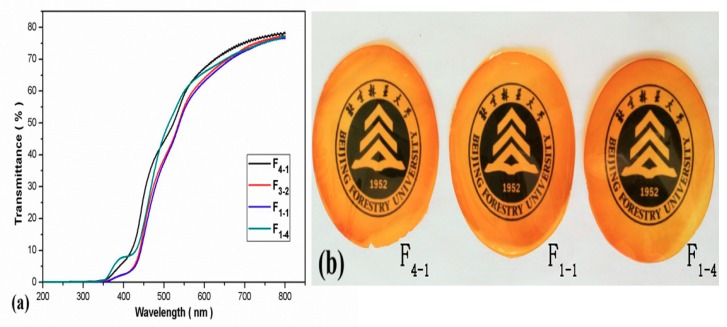
(**a**) UV-Vis transmittance of composite films (F_4-1_, F_3-2_, F_1-1_, F_1-4_); (**b**) Photograph of composite films (F_4-1_, F_1-1_, F_1-4_).

**Figure 6 materials-11-02264-f006:**
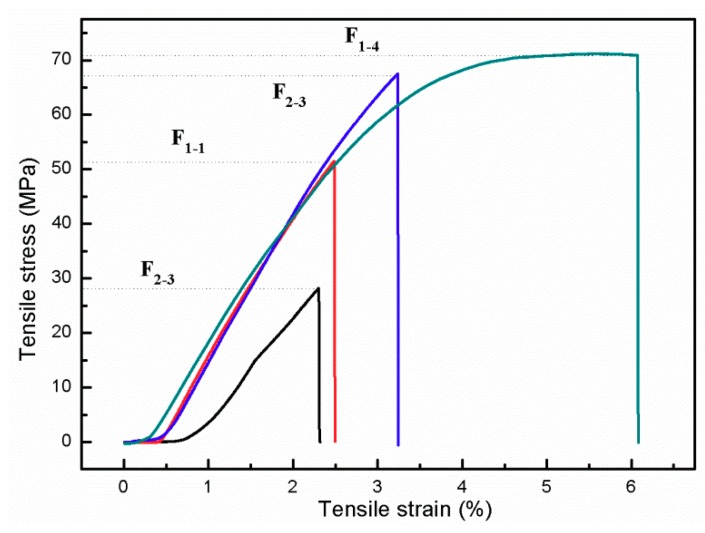
Tensile-strain curves of the composite films.

**Figure 7 materials-11-02264-f007:**
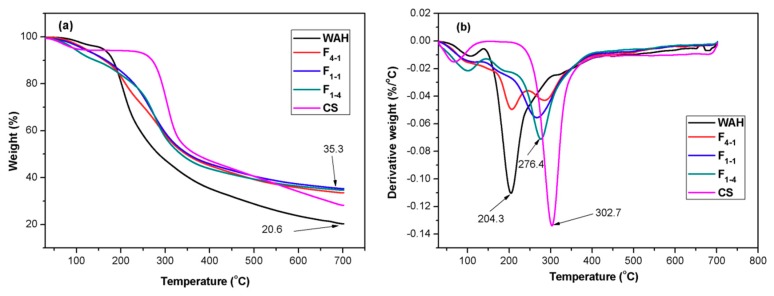
(**a**) The thermogravimetric analysis (TGA) curves of WAH, CS and composite films (F_4-1_, F_1-1_, F_1-4_), (**b**) the DTG curves of WAH, CS and composite films (F_4-1_, F_1-1_, F_1-4_).

**Table 1 materials-11-02264-t001:** Composite films with different ratios in volume of WAH and chitosan (CS).

Sample Name	WH (2 wt % *v*/*v*)	Chitosan (2 wt % *v*/*v*)
F_4-1_	80	20
F_3-2_	60	40
F_1-1_	50	50
F_2-3_	40	60
F_1-4_	20	80

**Table 2 materials-11-02264-t002:** Tensile testing results of the composite films.

**Sample**	**Tensile Strength (MPa)**	**Tensile Strain at Break (%)**		**Thickness (μm)**
F_3-2_	28.2 ± 1.3	2.3 ± 0.1		43.1 ± 3.0
F_1-1_	49.5 ± 1.8	2.5 ± 0.2		42.9 ± 2.0
F_2-3_	67.5 ± 2.0	3.2 ± 0.2		45.5 ± 3.0
F_1-4_	71.2 ± 1.5	6.1 ± 0.1		50.5 ± 2.0
**Films Reported in Literature**				
**Major Component (Reference)**	**Additional Components % (*w*/*w*)**	**Thickness (μm)**	**Tensile Strength (MPa)**	**Tensile Strain (%)**
Xylan [15]		290–380	1.1–1.4	45.6–56.8
Arabinoxylan [25]	2.7–20 glycerol	22–28	9.7–46.5	5.6–12.1
Chitosan [26]	50–70D-mannan	–	50–60	–

**Table 3 materials-11-02264-t003:** Thermal characteristics of TGA curves in Figure 7.

Curve	WAH	CS	F_4-1_	F_1-1_	F_1-4_
*T*_onset_ (°C)	158.1	243.2	183.4	204.2	237.6
*T*_max_ (°C)	204.3	302.7	284.7	270.4	276.4
Residual (wt %) at 700 °C	20.6	27.9	33.2	35.3	34.8

**Table 4 materials-11-02264-t004:** Oxygen transfer rate (OTR) and water vapor transmission rate (WVP) of the composite films and the films reported in literatures.

**Sample**	**OTR (cm^3^·m^−2^·24 h^−1^·kPa^−1^)**	**WVP (×10^−10^ g·cm·cm^−2^·s^−1^·mmHg^−1^)**	**Test Area (cm^2^)**
F_3-2_	0.34 ± 0.05	2.42 ± 0.33	5.0
F_1-1_	0.16 ± 0.01	2.17 ± 0.24	5.0
F_2-3_	0.30 ± 0.06	2.28 ± 0.19	5.0
F_1-4_	0.29 ± 0.05	3.82 ± 0.36	5.0
**Films Reported in Literatures**			
**Major Component (References)**	**Additional Components % (*w*/*w*)**	**Average Thickness (μm)**	**OTR (cm^3^·m^−2^·24 h^−1^·kPa^−1^)**
Arabinoxylan [27]	40 sorbitol	20-50	4.7
Polylactic acid Figurefilm [28]	–	25	18.65
AcGGM [29]	35 CMC	30–60	1.28

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
