# Peer review of "Composite Film Based on Pulping Industry Waste and Chitosan for Food Packaging"

_materials, 2018, doi:10.3390/ma11112264_

Round 1
Reviewer 1 Report
The topic of the paper is interesting and attractive and the results of concern; however, the paper should be revised before publication. This study and its results demonstrate an interesting potential application of composite film based on pulping industry waste and chitosan for food packaging. In this research, wood auto-hydrolysates and chitosan were used to construct wood auto-hydrolysate based films by the casting method. Additionally, properties of the composite films were described. The subject matter is interesting and potentially useful to readers of Materials. However, before it is suitable for publication, some opportunities for a major revision are:
1) Materials and methods
- Page 3, line 95: Please provide RH (%) of the drying chamber.
2) Results and Discussion
- Page 6, lines: 193-197: In addition, it might be due to the increment of the characteristic peak area of the films that the reaction intensities of WAH/CS increased with the increment of the WAH/CS content ratio from 4:1 (F4-1) to 1:1 (F1-1) to 2:3 (F2-3) (Fig. 2b). However, the characteristic peak area of the film F1-4 is smaller, which might be due to the relatively excessive content of CS. These results will be further discussed in the further discussion.
· The statement should be confirmed by the literature data.
· What did the authors mean with using the sentence: „These results will be further discussed in the further discussion”?
- Page 7, lines 221-222: This is consistent with the SEM results, and the smooth surfaces of the films were beneficial to the application of composite films in food packaging materials.
· Please, justify and explain in the manuscript why it is beneficial.
- Pages 6-8, sections: Morphology of Composite Films and UV-Vis transmittance of the Composite Films should be more discussed with other scientific reports.
- Tensile testing results of the composite films, Oxygen transfer rate (OTR) results and water vapor transmission rate (WVP) results of the composite films should be subjected to statistical analysis to see whether the changes are statistically significant.
- Page 10, lines 281-282: Oxygen transfer rate (OTR) and water vapor permeability (WVP) should be as low as possible in order to optimize the applications of composite films in food packaging.
This statement should be more precise and confirmed by the literature findings (focus on explaining why it is relevant)
- Page 10, lines 287-292: – Are below statements the authors' conclusions or statement from literature reports?
It might be due to the excess CS or WAH. Compared with the films reported in literatures, [26,27] the OTR values of composite films based on WAH and CS were relatively lower than those of the films, such as 1.28 cm3·m-2·24 h-1·kPa-1 reported for acetylated galactoglucomannans (AcGGM) and 18.65 cm3·m-2·24h-1·kPa-1 reported for polylactic acid film. Therefore, the WAH/CS films were suitable for the application in the food packaging. The WVP value of the composite films is an essential property for the food packaging. The WVP value of the composite films is an essential property for the food packaging materials. The WVP value of the films F1-1 was much lower than those of the film F3-2, F2-3, and F1-4, which might be due to the increment of the stronger reaction between WAH and CS.
- Page 10, lines 291 and 294-296: In addition, the WVP value of the film F3-2 was lower than those of F2-3 and F1-4. These lower WVP values demonstrated that the films based on WAH and CS are suitable for the application in food packaging.
· Please avoid the repetition in the manuscript
Author Response
Response to the reviews’ comments
Journal: Materials
Manuscript ID: materials-385219
Title: Composite film based on pulping industry waste and chitosan for food packaging
General response: We sincerely thank the editor and all reviewers for their valuable feedback that we have used to improve the quality of our manuscript. The reviewer comments are laid out below in italicized font and specific concerns have been numbered. Our response is given in normal blue font and changes/additions to the manuscript are given in red text.
Point-to-point response:
Reviewer 1
1) Materials and methods
- Page 3, line 95: Please provide RH (%) of the drying chamber.
Response: Thank you again for your positive comments and valuable suggestions to improve the quality of our manuscript. The film was obtained by heating the blended solutions at 50 °C in the vacuum drying chamber. The desiccant was placed in the vacuum drying chamber for absorbing the water, and the relative humidity could not be obtained in the chamber.
2) Results and Discussion
- Page 6, lines: 193-197: In addition, it might be due to the increment of the characteristic peak area of the films that the reaction intensities of WAH/CS increased with the increment of the WAH/CS content ratio from 4:1 (F4-1) to 1:1 (F1-1) to 2:3 (F2-3) (Fig. 2b). However, the characteristic peak area of the film F1-4 is smaller, which might be due to the relatively excessive content of CS. These results will be further discussed in the further discussion.
· The statement should be confirmed by the literature data.
· What did the authors mean with using the sentence: „These results will be further discussed in the further discussion”?
Response: Thank you for your positive comment and suggestion. The sentences “In addition, it might be due to the increment of the characteristic peak area of the films that the reaction intensities of WAH/CS increased with the increment of the WAH/CS content ratio from 4:1 (F4-1) to 1:1 (F1-1) to 2:3 (F2-3) (Fig. 2b). However, the characteristic peak area of the film F1-4 is smaller, which might be due to the relatively excessive content of CS. These results will be further discussed in the further discussion.” have been deleted.
- Page 7, lines 221-222: This is consistent with the SEM results, and the smooth surfaces of the films were beneficial to the application of composite films in food packaging materials.
· Please, justify and explain in the manuscript why it is beneficial.
Response: Thank you for your positive comment. The advantages of the smooth surfaces are described as follows: firstly, the smooth surface makes the packaging film more beautiful, then the inspection items of food packaging films in China includes: flat appearance, no scratch, no bubble, no wrinkle, and so on.
- Pages 6-8, sections: Morphology of Composite Films and UV-Vis transmittance of the Composite Films should be more discussed with other scientific reports.
Response: Thank you for your positive comment and suggestion. Some related modifications and supplements have been given in the manuscript in red.
- Tensile testing results of the composite films, Oxygen transfer rate (OTR) results and water vapor transmission rate (WVP) results of the composite films should be subjected to statistical analysis to see whether the changes are statistically significant.
Response: Thank you for your positive comment. The OTR and WVP of the composite films in the experiments were measured for at least 3-5 samples, and the results are based on at least 3 specimens. And this condition has been added in the section 2.9 and 2.10 of the manuscript.
- Page 10, lines 281-282: Oxygen transfer rate (OTR) and water vapor permeability (WVP) should be as low as possible in order to optimize the applications of composite films in food packaging.
This statement should be more precise and confirmed by the literature findings (focus on explaining why it is relevant)
Response: Thank you for your positive comment and suggestion. The precise OTR standard of food packaging films has been added in the manuscript. The sentence “In addition, the standard maximum OTR value of food packaging materials is below 10 cm3·m-2·24 h-1·kPa-1 [28]” have been added in the manuscript.
- Page 10, lines 287-292: – Are below statements the authors' conclusions or statement from literature reports?
It might be due to the excess CS or WAH. Compared with the films reported in literatures, [26,27] the OTR values of composite films based on WAH and CS were relatively lower than those of the films, such as 1.28 cm3·m-2·24 h-1·kPa-1 reported for acetylated galactoglucomannans (AcGGM) and 18.65 cm3·m-2·24h-1·kPa-1 reported for polylactic acid film. Therefore, the WAH/CS films were suitable for the application in the food packaging. The WVP value of the composite films is an essential property for the food packaging. The WVP value of the composite films is an essential property for the food packaging materials. The WVP value of the films F1-1 was much lower than those of the film F3-2, F2-3, and F1-4, which might be due to the increment of the stronger reaction between WAH and CS.
Response: Thank you for your positive question, the related statement was the conclusions drawn by the author through comparing the experimental data with the data in the literature. And some expressions have been modified in the manuscript.
- Page 10, lines 291 and 294-296: In addition, the WVP value of the film F3-2 was lower than those of F2-3 and F1-4. These lower WVP values demonstrated that the films based on WAH and CS are suitable for the application in food packaging.
Please avoid the repetition in the manuscript
Response: Thank you for your positive comment, some modifications have been made in the manuscript.
Reviewer 2 Report
The manuscript is prepared according with Journal Materials. The presented and studied topic has a good scientific significance and novelty. Potentially renewable materials such as wood auto-hydrolysates are presented as alternative for future food packaging applications.
Overall paper is clear presented, with suggestive abstract and conclusions. The highlights are related to the core results of the paper. The evaluation methods and experimental techniques are relevant, obtained results are credible. In manuscript, relevant references are cited.
Although, here are some questions and remarks:
1) Is Figure 1 authors work?
2) Line 107-114; Table 1: spaces and lines should be corrected.
3) Chapter 2.7. Tensile Strength Testing: line 144: Authors explain that the specimens were kept at room temperature in drying oven for 3 days. Was the humidity in the oven also measured?
4) Line 166-168 where the equation of WVP is presented: correct that all equation will be in 1 line.
5) Line 298: Table 4: Major component (References): 25, 26 and 27 should be in bracket [25]…What does it mean 40 sorbitol; 35 CMC?
Author Response
Response to the reviews’ comments
Journal: Materials
Manuscript ID: materials-385219
Title: Composite film based on pulping industry waste and chitosan for food packaging
General response: We sincerely thank the editor and all reviewers for their valuable feedback that we have used to improve the quality of our manuscript. The reviewer comments are laid out below in italicized font and specific concerns have been numbered. Our response is given in normal blue font and changes/additions to the manuscript are given in red text.
Point-to-point response:
Reviewer 2
1) Is Figure 1 authors work?
Response: Thank you for your positive comment, and the image was done by me.
2) Line 107-114; Table 1: spaces and lines should be corrected.
Response: Thank you again for your positive suggestion, the table was made corresponding modifications in the manuscript.
3) Chapter 2.7. Tensile Strength Testing: line 144: Authors explain that the specimens were kept at room temperature in drying oven for 3 days. Was the humidity in the oven also measured?
Response: Thank you again for your positive question. Saturated Mg(NO3)2 solution was placed in the cabinet, and the relative humidity of saturated Mg(NO3)2 solution was about 50% at room temperature, so the humidity in the cabinet was stable.
4) Line 166-168 where the equation of WVP is presented: correct that all equation will be in 1 line.
Response: Thank you again for your positive suggestion, the equation has been corrected in 1 line.
5) Line 298: Table 4: Major component (References): 25, 26 and 27 should be in bracket [25]…What does it mean 40 sorbitol; 35 CMC?
Response: Thank you again for your positive suggestion, the “25”, “26” and “27” have been corrected to [25], [26], and [27] in the manuscript.
And “40 sorbitol; 35 CMC” mean “additional components of 40% (w/w) sorbitol; additional components of 35% (w/w) CMC”. The corresponding modification was made in Table 4 in the manuscript.
Round 2
Reviewer 1 Report
Authors made the changes as was highlighted in the review report, therefore I recommend to accept the manuscript in the present (updated) form.